# Teacher Perception of Automatically Extracted Grammar Concepts for L2 Language Learning

**Aditi Chaudhary**[†][*], **Arun Sampath**[△], **Ashwin Sheshadri**[△],
**Antonios Anastasopoulos**[‡], **Graham Neubig**[†]
[†]Carnegie Mellon University, [△]Kannada Academy, [‡]George Mason University
aditichaud@google.com    gneubig@cs.cmu.edu    antonis@gmu.edu
{arun.sampath,ashwin.sheshadri}@kannadaacademy.com

## Abstract

One of the challenges in language teaching is how best to organize rules regarding syntax, semantics, or phonology in a meaningful manner. This not only requires content creators to have pedagogical skills, but also have that language's deep understanding. While comprehensive materials to develop such curricula are available in English and some broadly spoken languages, for many other languages, teachers need to manually create them in response to their students' needs. This is challenging because i) it requires that such experts be accessible and have the necessary resources, and ii) describing all the intricacies of a language is time-consuming and prone to omission. In this work, we aim to facilitate this process by *automatically* discovering and visualizing grammar descriptions. We extract descriptions from a natural text corpus that answer questions about morphosyntax (learning of *word order, agreement, case marking, or word formation*) and semantics (learning of *vocabulary*). We apply this method for teaching two Indian languages, Kannada and Marathi, which, unlike English, do not have well-developed resources for second language learning. To assess the perceived utility of the extracted material, we enlist the help of language educators from schools in North America to perform a manual evaluation, who find the materials have potential to be used for their lesson preparation and learner evaluation.

## 1 Introduction

Recently, computer-assisted learning systems have gained tremendous popularity, especially during the COVID-19 pandemic when in-person instruction was not possible, leading to the need for user-friendly and accessible learning resources (Li and Lalani, 2020). Because these materials are curated by subject experts, this makes curriculum design a challenging process, especially for languages where experts or resources are inaccessible.

In the language learning context this entails designing materials for different learning levels, covering different grammar points, finding relevant examples, and even creating evaluation exercises. For second language (L2) learning, it is not straightforward to reuse existing curricula even in the same language, as the requirements of L2 learners could be vastly different from the traditional first language (L1) setting (Munby, 1981). Given that only a handful of languages, in particular English, have a plethora of resources for L2 learning, but for most of the world's 4000+ written languages (Eberhard et al., 2022), it is a struggle to find even a sufficiently large and good quality text corpus (Kreutzer et al., 2021), let alone teaching material. In this paper, we explore *to what extent can a combination of NLP techniques and corpus linguistics assist language education for languages with limited teaching as well as text resources?*

With technology advancements, teachers have used corpus-based methods (Yoon, 2005) to analyze large text corpora and find patterns such as collocations and relevant examples of language use, (Davies, 2008; Cobb, 2002), to supplement their vocabulary teaching. Now, with advances in NLP, we can extract instructional material for more complex use cases. For instance, the popular tasks of Part-of-Speech (POS) tagging and dependency parsing do answer questions about some *local* aspects of the language such as 'what is the function of words or their relations'. AutoLEX (Chaudhary et al., 2022) uses this local analysis for extracting answers to linguistic questions in both human- and machine-readable format. Given a question (e.g. "how are objects ordered with respect to verbs in English"), AutoLEX formalizes it into an NLP task and learns an algorithm which extracts not only the common patterns (e.g. "object is before/after the verb") but also the conditions which trigger each of them (e.g. "objects come after verbs except for interrogatives").

---

[*]Currently works at Google Research

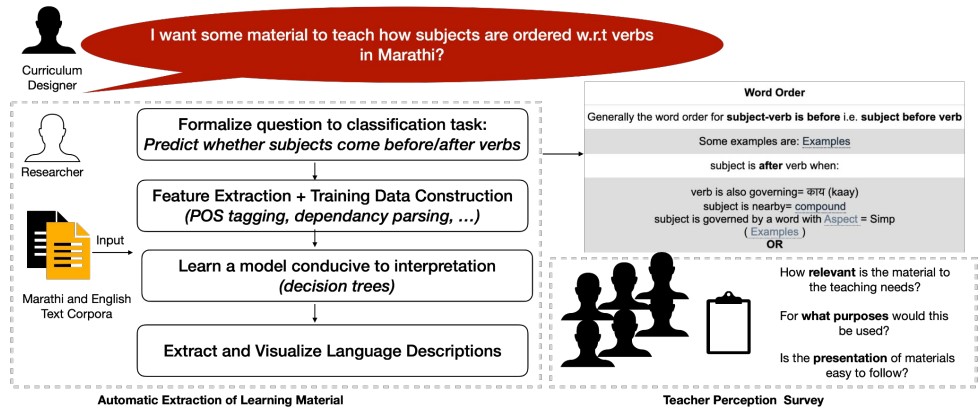

Figure 1: NLP researchers work with curriculum designers to understand their teaching needs. We then formulate an NLP task to learn a model from which we can extract and visualize learning material. Finally, we work with in-service teachers to understand their perception of the extracted materials for relevance, utility and presentation.

In this paper, we take a step further by examining the utility of such extracted descriptions for language education. We collaborate with in-service teachers, where we tailor automatically extracted grammar points to their teaching objectives. The process starts with identifying "teachable grammar points" which are individual syntactic or semantic concepts that can be taught to a learner. For example, with respect to *word order*, a teachable grammar point could be understanding "how subjects are positioned with respect to verbs". We apply AutoLEX to extract human-readable explanations directly from the text corpora of the language of interest. Finally, we present the extracted materials to in-service teachers to evaluate the utility in their teaching process. Figure 1 outlines this collaborative design. To our knowledge, use of such automatically extracted insights have not been explored for language teaching (see section 7 for related work). We summarize our contributions below –

- We explore how NLP methods can play a role in language pedagogy, especially for languages which lack resources. This entails designing teaching points by including real users (teachers in our case) in a collaborative design process and further evaluating its its practical utility with a large human study.

- We automatically extract leaning materials for Kannada and Marathi, two Indian languages, and present them through an online interface[1] and release the code/data publicly.[2]

- We conduct a survey with 17 in-service teachers to understand their perception of the extracted materials – 85% Kannada teachers and 40% Marathi teachers note that they will likely use this material for lesson preparation, and for providing additional material to students for self-exploration.

## 2 Why Marathi and Kannada?

Although these languages are spoken primarily in India, a small but significant populace of speakers has emigrated, resulting in a demand to maintain language skills within this diaspora. We identified schools in North America that teach these languages to English speakers including children and adults, with their primary objective to a) preserve and promote the language and culture and, b) help speakers communicate with their community. While there are existing textbooks, they cannot be used as-is, as they are based on more *traditional L1 teaching* (Selvi and Shehadeh, 2018), where the language is taught from the ground up, from introducing the alphabet, its pronunciation and writing, to each subsequent grammar point. Teachers have instead adapted the existing material and continue to design new material to suit the L2 speakers' needs. Additionally, in comparison to English, both these languages have far fewer L2 learning resources. Both Marathi and Kannada are not part of any popular tools (e.g. Duolingo[3] or Rosetta Stone (Stone, 2010)). For Marathi, there is an online learning tool Barakhadi[4] which, however, is not

---

[1] https://www.autolex.co
[2] https://github.com/reviewfornlp/

teacher-perception
[3] https://www.duolingo.com/
[4] https://barakhadi.com/

free of cost. Therefore, these languages are under-resourced with respect to pedagogical resources, and will likely benefit from this exercise.

## 3 Proposed Work

Although language education has been widely studied in literature, there is no one 'right' method of teaching. Since our focus is to create pedagogical content to assist teachers in their process, we conduct a pilot study and collaborative design with two Kannada teachers who are deeply involved in the curriculum designing. We first identify some "teachable grammar points" which, as defined above, are points that can be taught to a learner and are typically included in a learning curriculum. Next, following AutoLEX, we formulate each grammar point into an NLP task and extract human-readable descriptions, as shown below.

### 3.1 Identify Teachable Grammar Points

As a first step towards the curriculum design scenario, we performed an inventory of the aspects taught in existing teaching materials. We manually inspected three of the eight Kannada textbooks shared by the curriculum designers, which are organized by increasing learning complexity, and identified grammar aspects such as identification of word categories (e.g. nouns, verbs, etc.), vocabulary, and suffixes. In Table 1, we show examples of the teachable grammar points that we attempt to extract, which we group under five grammar aspects namely General Information, Vocabulary, Word Order, Suffix Usage and Agreement. We only cover the above subset out of all grammar aspects described in the textbooks because they satisfied two desiderata: a) the Kannada teachers identified these to be widely studied and important in their curriculum and b) the underlying linguistic questions can be formulated into an NLP task thereby allowing us to extract descriptions.

### 3.2 Extract Learning Material

As noted above, we build on AutoLEX (Chaudhary et al., 2022) to extract learning material. AutoLEX takes as input a raw text corpus, comprising of sentences in the language of interest, and produces human- and machine-readable explanations of different linguistic behaviors, following a four step process. The first step is formulating a linguistic question into an NLP task. For example, given a question "how are objects ordered with respect to verbs in English", it is formalized it into an NLP task "predict whether the object comes before or after the verb". The second step is to learn a model for this prediction task, for which training data is constructed by identifying and extracting features from the text corpora that are known to govern the said phenomenon (e.g. POS tagging and dependency parsing). Next, AutoLEX learn an algorithm which extracts not only the common patterns (e.g. "object is before/after the verb") but also the conditions which trigger each of them (e.g. "objects come after verbs except for interrogatives"). Importantly, for each pattern, illustrative examples with examples of exceptions are extracted from the corpora. Finally, the extracted conditions are visualized with illustrative examples through an online interface. We adapt the above process to extract descriptions for all the teachable grammar points defined in Table 1. Of those, AutoLEX already outlines the process for agreement and word order and for the others we adapt the process as shown below.

**Word Order and Agreement** Both Marathi and Kannada are morphologically rich, with highly inflected words for gender, person, number; morphological agreement between words is also frequently observed. Both languages predominantly follow SOV word order, but because syntactic roles are often expressed through morphology, there are often deviations from this order. Therefore, learners must understand both the rules of word order and agreement to produce grammatically correct language. In AutoLEX, word order and agreement grammar points are extracted by formulating questions, as shown in Table 1, and learning a model to answer each question. As outlined above, to train these models, we must first identify the relevant elements (e.g. for word order, subjects, and verbs) to construct the training data. To do so, the corpus of that language must be syntactically annotated with POS tags, morphological analyses, lemmas, and dependency parses. Next, to discover when the subject is before or after, AutoLEX extracts syntactic and lexical signals from other words in that sentence and uses them to train a classifier. To obtain interpretable patterns, we use decision trees (Quinlan, 1986), similar to AutoLEX, which extract "if X then Y" style patterns that can, if presented appropriately, be interpreted by teachers or learners. Example word order patterns extracted from this model for Marathi are shown in Figure 1. While AutoLEX uses English as the meta-language to present ex-

| Aspects | Teachable Grammar Points |
|---|---|
| General Information | What gender values does Marathi show? (e.g. masculine, feminine, neuter)
Which type of words show these values? (e.g. nouns, verbs)
What are some example word usages? |
| Vocabulary | What words to use for popular categories (e.g. food, animals, etc.)
What are some adjectives, their synonyms and antonyms?
Which word to use when? |
| Word Order | Are subjects before or after verbs in Marathi?
If both, when is subject before and when is it after? |
| Suffix Usage | What are the common suffixes for Marathi nouns?
When should a particular suffix (e.g. -'laa') be used? |
| Agreement | Do some words need to agree on gender?
If so, when should they necessarily agree and when they need not? |

Table 1: Example aspects of grammar, vocabulary and teachable grammar points covered in our material.

tracted descriptions, we use a combination of L1 (English) and L2. This is in alignment with methods such as *Grammar-Translation* (Doggett, 1986) that encourage learning using both L1 and L2.

**Suffix Usage** Along with understanding sentence structure, it is equally important to understand how inflection works at the word level. The first step is to identify the common suffixes for each word type (e.g. nouns) and then ask "which suffix to use when". For that, we need to identify the POS tags and produce a morphological analysis for each word, like we did above. To identify the suffix, we train a model that takes as input a word with its morphological analysis (e.g. 'deshaala,N,Acc,Masc,Sing') and outputs the decomposition (e.g. 'desh + laa'). Next, a classification model is trained for each such suffix (e.g. '-laa') to extract the conditions under which one suffix is typically used over another (Figure 2).

**Vocabulary** Vocabulary is probably one of the most important components of language learning (Folse, 2004). There are several debates on the best strategy for teaching vocabulary; we follow prior literature (Groot, 2000; Nation, 2005; Richards et al., 1999), which use a mixture of definitions with examples of word usage in context. Specifically, we organize the material around three questions, as shown in Table 1. There are some categories of words where the same L1 (English) word can have multiple L2 translations, with fine-grained *semantic subdivisions* (e.g. 'bhaat' and 'tandul' both refer to 'rice' in Marathi, but the latter refers to raw rice and the former refers to cooked rice). Chaudhary et al. (2021) propose a method for iden-

tifying such word pairs, along with explanations on their usage, using parallel sentences between English and the L2. Each pair of sentence translations is first run through an automatic word aligner (Dou and Neubig, 2021), which extracts word-by-word translations, producing a list of English words with their corresponding L2 translations. On top of this initial list, filtration steps are applied to extract those word pairs that show fine-grained divergences. Training data is then constructed to solve the task of lexical selection, i.e. for a given L1 word (e.g. 'rice') in which contexts to use one L2 word over another (e.g. 'bhaat' vs 'tandul'). Because most of our learners have English as their L1, we extract signals from both the L1 and L2 corpora to train the classifier and thereby derive style patterns which contain both L1 and L2. *Communicative Approach* (Johnson and Brumfit, 1979) focuses on teaching through functions (e.g. self-introduction, identification of relationships, etc.) over grammar forms; therefore, we also organize vocabulary around popular categories. We run a word-sense disambiguation (WSD) model (Pasini et al., 2021) on English sentences, which helps us to identify the word sense for each word in context (e.g. 'bank.n.02' refers to a financial institution while 'bank.n.01' refers to a river edge). Given the hierarchy of word senses expressed in WordNet (Miller, 1995), we can traverse the ancestors of each sense to find whether it belongs to any of the pre-defined categories (e.g. food, relationships, animals, fruits, colors, time, verbs, body parts, vehicle, elements, furniture, clothing). Example Marathi words extracted are shown in Figure 4. We also identify popular adjectives, their synonyms,

and antonyms, also extracted from WordNet, and present them in a similar format (Figure 5).[5] For each word, we also present accompanying examples that illustrate its usage in context, along with its English translations. For the benefit of users who are not familiar with the script of L2 languages, we automatically transliterate into Roman script using Bhat et al. (2015).

**General Information** In addition to answering these morpho-syntax and semantic questions, we also present salient morphology properties at the language level. Specifically, from the syntactically parsed corpus of the target language, we hope to answer basic questions such as "what morphological properties (e.g. gender, person, number, tense, case) does this language have", as shown in Table 1. These questions were inspired from the Kannada textbooks shared by experts, which introduces the learner to basic syntax and morphology. Understanding syntax patterns are crucial, especially for Kannada and Marathi, which show significant variations in inflection. Through the previous vocabulary section, learners can learn the L2 words for action verbs, and through this section, they can learn how to use those verbs for different genders, tenses, etc. For each question, we organize the information by frequency, a common practice in language teaching where textbooks often comprise of frequently used examples (Dash, 2008).

Along with relevant content, the format in which the material is presented is equally important. Smith Jr (1981) outline four steps involved in language teaching: *presentation*, *explanation*, *repetition* of material until it is learned, and *transfer* of materials in different contexts, which have no fixed order. For example, some teachers prefer the presentation of content (e.g. reading material, examples, etc.) first followed by explanation (e.g. grammar rules), while Smith Jr (1981) discuss that, for above-average learners, explanation followed by presentation may be preferable. In our design, we extract and present both (i.e. rules and examples) without any specific ordering, allowing educators to decide based on their requirements. By providing illustrative examples from the underlying text at each step, we hope to address the *transfer* step, where learners are exposed to real language .

---

[5]First, we automatically identify frequent cross-lingual word pairs from our corpus. Next, we identify the adjectives using POS tagging and use WordNet to extract the antonyms/synonyms of their English counterparts.

## 4 Automatic Evaluation

Since our objective is to evaluate whether such automatically derived linguistic insights can be useful for language pedagogy, we first conduct a pilot study to evaluate *quality* and *properties* of the extracted materials (section 5). Next, we conduct a study with several in-service teachers, both in Kannada and Marathi, to evaluate *relevance*, *usability*, and *presentation* of the extracted materials (section 6). In addition to human evaluation, we follow Chaudhary et al. (2022) to automatically evaluate the quality of extracted descriptions. This provides a quick sanity check on whether our trained models are able to learn the said linguistic phenomena.

**Word order and Agreement** Chaudhary et al. (2022) automatically evaluate the learnt model by measuring the accuracy on held-out sentences. For example, for subject-verb model, the gold label is the observed word order which can be determined from the POS and dependency parses (i.e. whether the subject is 'before' or 'after' the verb), which is then compared with the model prediction to compute the accuracy. This model is compared with a baseline that assigns the most frequent observed pattern in the training data as model prediction.

**Suffix Usage** We use a similar most-frequent baseline where the most frequent suffix pattern is used as model prediction for the baseline accuracy, where the observed suffix is the gold label.

**Vocabulary** We follow the accuracy computation from Chaudhary et al. (2021) to evaluate the model used for extracting semantic subdivisions – for each sentence the model prediction is compared with the gold label which is the observed L2 word for the L1 word. The baseline uses the most frequently observed L2 word translation for the given L1 word and is compared with the gold label to compute the baseline accuracy.

### 4.1 Setup

**Data** Since our goal is to create teaching material for learners having English as L1, we use the parallel corpus of Kannada-English and Marathi-English from SAMANANTAR (Ramesh et al., 2022) comprising of 4 million sentences, as our starting point. This covers text from a variety of domains such as news, Wikipedia, talks, religious text, movies.

**Model** As mentioned in section 3, the first step in the extraction of materials is to parse sentences for POS tags, morphological analysis and dependency parsing. To obtain this analysis for our corpus,

we use UDIFY (Kondratyuk and Straka, 2019) that jointly predicts POS tags, lemma, morphology and dependency tree over raw sentences. However, UD­IFY requires training data in the UD annotation scheme (McDonald et al., 2013). Kannada has no UD treebank available and for Marathi the tree­bank is extremely low-resourced covering only 300 sentences. Therefore, we train our own parser as outlined in subsection A.1. To learn models for ex­tracting descriptions, we follow the same modeling setup as Chaudhary et al. (2021) and Chaudhary et al. (2022) and use decision trees (Quinlan, 1986) to extract the patterns, explanations and accompa­nying examples (subsection A.2). For suffix usage, we additionally train a morphology decomposition model (Ruzsics et al., 2021) which breaks a word into its lemma and suffixes, over which we learn a classification model.

## 4.2 Results

In Table 5 we report results for word order, suffix usage and agreement. We can see that in most cases, the rules extracted by the model outperform the respective baselines, suggesting that the model is able to extract decent first-pass rules, with 98% avg. accuracy for Kannada word order, 48% for agree­ment, 85% for suffix usage, 68% for vocabulary, 98% for Marathi word order, 61% for agreement, 85% for suffix usage and 70% for vocabulary.[6]

## 5 Human Quality Evaluation

We conduct a limited study for a sanity check with two Kannada teachers.

**Vocabulary**   We present both experts with an au­tomatically generated list of 100 English-Kannada word pairs, where one English word has multiple translations showing fine-grained semantic sub­divisions. Both experts found 80% of the word pairs to be valid, according to the criterion that they show different usages. For example, for 'doc­tor', the model discovered four unique translations, namely 'vaidya, vaidyaro, daktor, vaidyaru' in Kannada which the expert found interesting for teaching as they demonstrated fine-grained diver­gences, both semantically and syntactically. For instance, 'vadiya' is the direct translation of 'doc­tor', whereas 'daktor' is the English word used

as-is, 'vaidyaro' is the plural form and 'vaidayaru' is a formal way of saying a doctor.

**Word Order**   For word order, experts evaluate the rules for subject-verb and object-verb order. For subject-verb, seven grammar rules explaining the different word order patterns were extracted (4 explaining when the subject can occur both before and after the verb, 2 rules informing when sub­jects occur after, and 1 showing the default order of "before"). Of the seven rules, experts found four to be valid patterns. For object-verb word order, of the six rules extracted by the model, experts marked that 2 rules precisely captured the patterns, while one rule was too fine-grained. Interestingly, in addition to correctly identifying the dominant or­der, all the rules which were deemed valid showed non-dominant patterns. The rules marked as in­valid were invalid because the syntactic parser that generated the underlying syntactic analyses incor­rectly identified the subjects/objects. Such errors are expected given that there is not sufficient quan­tity/quality of expertly annotated Kannada syntac­tic analyses available to train a high-quality parser. However, we would argue that these results are still encouraging because i) despite imperfect syntactic analysis, the proposed method was able to extract several interesting counter-examples to the domi­nant word order, and ii) further improvements in the underlying parsers for low-resource languages may be expected through active research.

**Suffix Usage**   We extract different suffixes used for each word type (e.g. nouns, adjectives, etc.) but in the interest of time asked experts to evaluate only the suffixes extracted for nouns and verbs. Of the 18 noun suffixes, 7 were marked as valid, 2 suffixes were not suffixes in traditional terms but arise due to "sandhi" i.e. transformation in the characters at morpheme boundaries. Similarly, for verb suffixes, 53% (7/13) were marked as valid. Experts mentioned that understanding suffix usage is particularly important in Kannada, as it is an agglutinative language with different affixes for different grammar categories.

## 6 Teacher Perception Study

For Kannada, we work with teachers from the Kan­nada Academy[7] (KA), which is one of the largest organizations of free Kannada teaching schools in the world and recruit 12 volunteer teachers. For

---

[6]Because there tends to be strong agreement in these lan­guages, there is a class imbalance which probably led to the low performance of the agreement classifier.

[7]https://www.kannadaacademy.com/

Marathi, there is no central organization as for Kannada, but there are many independent schools in North America. We reached out to Marathi Vidyalay[8] in New Jersey, that teaches learners in the age group of 6-15, and Shala in Pittsburgh[9]. Marathi Vidyalay is a small school consisting of seven volunteer teachers, of whom four agreed to participate, while Shala has one teacher. All participants are volunteer teachers; teaching is not their primary profession. Since we extract learning materials automatically from publicly available corpora which may contain material which is age-inappropriate, we purposefully chose to share these materials with the teachers who we feel are best suited to decide how to use them.

**Perception Survey** To answer the research question of whether materials are practically usable and, if so, with regard to what aspects, we analyze the Kannada and Marathi teachers' perception regarding *relevance*, *utility* and *presentation* of the materials. First, a 30–60 minute meeting is conducted for the teachers, in which we introduce the tool, the different grammar points covered in it, and how to navigate the online interface. Teachers have one week to explore the materials. Finally, all teachers receive a questionnaire Table 3 that requires them to assess the relevance, utility and presentation.[10]

### 6.1 Kannada Results and Discussion

We report individual results in Table 2. 12 teachers with varying levels of teaching experience participated in this study.[11] All teachers have used some online tools, but mostly for creating assignments for the learners (e.g. Google Classroom, Kahoot[12], Quizlet[13]), or conducting classes (e.g. Zoom). However, none had used immersive online tools similar to our tool.

**Relevance** We see that teachers, on average, find 45–60% of material presented as relevant to their

---

[8]https://marathivishwa.org/marathi-shala/
[9]https://www.mmpgh.org/MarathiShala.shtml

[10]Although we conducted a manual evaluation of a subset of extracted materials in section 5, we did not remove those items that were marked as incorrect by the experts as we wanted to understand how the materials, as directly obtained automatically without significant human intervention, are perceived when presented as-is. This is close to the real world setting where human evaluation of each grammar point is not feasible.

[11]Four teachers had less than three years of experience, four teachers between 3-10 years, and the remaining had 10+ years of experience. Four teachers teach only beginners, while others have experience teaching higher levels as well.

[12]https://kahoot.com/
[13]https://quizlet.com/

existing curriculum. This is notable given that the underlying corpus is not specifically curated for language teaching and contains rather formal language. All teachers noted that especially for beginners they prefer starting with simpler and more conversational language style, but 5 teachers explicitly mentioned that for advanced learners this would be very helpful. In fact, one of the teachers having 3+ years of experience teaching intermediate to advanced learners explicitly mentioned that–

> *"The examples are well written, however, for beginners and intermediates, this might be too detailed. The corpus could be from a wider data source. The use of legal terms is less commonly used in day-to-day life. Advanced learners will certainly benefit from this."*

**Utility** We find that for all grammar concepts, most teachers (nearly 80%) expressed that they were likely to use the materials for lesson preparation. The per-grammar category results are in Table 2. In fact, one teacher who used our materials to teach suffix usage to an adult learner said–

> *"I used this tool to teach an American adult who takes private lessons and found it helpful in addressing her grammar questions. I liked how it was clearly segregated i.e. the suffixes for nouns vs proper-nouns and how it is different from one another. It is definitely great tool to refer for adults but again the vocabulary is perfect to improve written skills than the spoken language".*

Some teachers also mentioned that they could present the material to students for self-exploration, and about 70% teachers noted that it would be especially helpful for vocabulary learning. When asked what aspects of the presented material they would consider using, all teachers said that they would use the illustrative examples for all sections except for the word order and agreement sections. For agreement and word order sections, although they liked the general concepts presented in the material (for example, the non-dominant patterns shown under each section), 88% of the teachers felt that the material covered advanced topics outside the current scope. Although the quality evaluation of the rules was not part of this study, teachers noted that if the accuracy of the rules, particularly for

| Grammar Concept | Relevance % of relevant curriculum covered | | % of teachers likely to use | | Utility % of teachers that would use for | | Presentation % of teachers found this ___ to navigate | |
|---|---|---|---|---|---|---|---|---|
| | ka | mr | ka | mr | ka | mr | ka | mr |
| General Information | 62.1% | 15% | highly likely: 8.3%
likely: **83.3%**
not likely: 8.3% | -
40%
**60%** | lesson prep:**1.8%**
student exploration: 54.5%
student evaluation: 10% | **100%**
50%
- | very easy: 33.3%
somewhat easy: **58.3%**
difficult: 8.3% | -
**80%**
20% |
| Vocabulary | 67.5% | 16% | highly likely: 33.3%
likely: **58.3%**
not likely: 8.3% | -
40%
**60%** | lesson prep: **72.7%**
student exploration: 72.7%
student evaluation: 45.5% | **100%**
50%
50% | very easy: 36.3%
somewhat easy: **58.3%**
difficult: 8.3% | -
**100%**
- |
| Suffix Usage | 52.5% | 9% | highly likely: 9.1%
likely: **72.7%**
not likely: 18.2% | -
40%
**60%** | lesson prep: **77.8%**
student exploration: 55.6%
student evaluation: 33.3% | **100%**
50%
- | very easy: 36.4%
somewhat easy: **63.6%**
difficult: 0% | -
**100%**
- |
| Word Order | 66% | 8% | highly likely: 10%
likely: **70%**
not likely: 20% | -
40%
**60%** | lesson prep: **88.9%**
student exploration: 44.2%
student evaluation: 22.2% | **100%**
50%
- | very easy: 27.3%
somewhat easy: **72.7%**
difficult: 0% | -
**80%**
20% |
| Agreement | 53.75% | 5% | highly likely: 20%
likely: **60%**
not likely: 20% | -
40%
**60%** | lesson prep: **77.8%**
student exploration: 44.4%
student evaluation: 22.4% | **100%**
50%
- | very easy: 36.4%
somewhat easy: **45.5%**
difficult: 18.2% | -
**80%**
20% |

Table 2: Perception study results from 12 Kannada and 5 Marathi teachers

suffix usage, could be further improved, they could foresee this tool being used in classroom teaching, as suffixes are essential in Kannada.

**Presentation**    In terms of presentation of the materials, all teachers found them easy to navigate through, although it took some getting used to. This is expected given that the teachers spent only a few hours (5-6) over the course of a week exploring all materials. 8 teachers noted that for a new user the materials could be overwhelming to navigate but for instance, the two Kannada experts, who also participated in the quality study, have had weeks of exposure to the tool and therefore rated it very easy to navigate. We also find that these results vary for different grammar categories covered, for example, for the Vocabulary section, generally the materials presented were 'somewhat-easy' to 'easy' to navigate, while for the Agreement section, 18.2% teachers found the materials difficult to navigate. One of the reasons could be the meta-language used to describe the materials, for instance for Agreement the rules consisted of formal linguistic jargon (for example, most teachers were unfamiliar with the term 'lemma' or the different POS tags)[14].

## 6.2   Marathi Results and Discussion

For Marathi, five teachers participated in the study, all of whom teach at the beginner level with a few intermediate learners.

**Relevance**    Teachers find only 10–15% of the materials as relevant to their existing curriculum. This

[14]For reference, we had created a documentation for the teachers in the study, which provides definition of these formal terms along with examples, but it is hard to determine whether the teachers consulted them frequently while evaluating.

is much less than what the Kannada teachers reported, probably because the Marathi schools' primary focus is teaching beginners. For beginners, teachers begin by introducing the alphabet, simple vocabulary, and sentences. In our tool, currently we do not curate the material according to learner age/experience, and we have extracted the learning materials from a public corpus which comprises of news articles that are not beginner-oriented.

**Utility**    Unlike in the Kannada findings, where 85% teachers said that were 'likely to use' the materials, for Marathi, 60% teachers said 'not likely to use'. The main reason being that the Marathi teachers mainly teach beginner levels and found the materials more suited for advanced learners. The teachers who marked that were 'likely to use' noted that they would use them for lesson preparation. 50% of those also said that they could provide the materials to advanced students for self-exploration, to encourage them to explore the materials and ask questions. Similarly to the Kannada study, two teachers found the illustrative examples to be of the most utility, as they demonstrate a variety of usage. However, they did note that because the underlying corpus was too restricted in genre, they would benefit more from applying this tool to their curated set of stories, which are age-appropriate.

**Presentation**    88% of the teachers found the materials 'somewhat easy' to navigate and similar to the Kannada teachers, mentioned that it did require some time to understand the format. The teachers also said that currently the material is too content heavy and not visually engaging, if the presentation could be improved along those aspects, it would

make the tool more inviting.

# 7 Related Work

Below, we discuss some relevant literature for language learning.

**Automatic Assessment** Automatically assessing a learner's progress is perhaps the most popular NLP application explored in the past. For instance, Zou et al. (2022) automatically generate true/false question to assess an English learner's reading comprehension. Wambsganss et al. (2022) provide feedback on erroneous argument structures to help improve an English learner's essay writing skills. While most work has been for English, some works have developed assessment tools for other languages, for example, Weiss and Meurers (2022) assess sentence readability for German L2 learners, Imperial et al. (2022) build the first readability model for Cebuano which assesses the readability level of children's books. Similar to us, they also use interpretable models (e.g. SVM, Random-Forests) trained using linguistic features extracted from a text corpora. However, their focus is on classifying the content into three learner levels, while our focus is towards extracting teaching content from the corpus.

**Educational Tools** Over the years there has been a surge in language learning tools such as Rosetta Stone (Stone, 2010), Duolingo[15], LingQ[16], LearnALanguage[17], Omniglot[18]. Most of these tools have learning content manually curated with the help of subject matter experts, which, however, makes it difficult to extend them to numerous languages. Recently, NLP tools have been used to develop resources for low-resource languages, for instance, Ahumada et al. (2022) use a combination of linguistic resources (e.g. grammars), NLP tools (e.g.- morphological analyzers) and community resources (e.g. dictionaries) to build learning tools for the indigenous language Mapuzugun. Specifically, they design an orthography recognizer which identifies which of the three alphabets the input text is in, converts across orthographies if required, performs word segmentation and analysis and maps to them user-readable phrases/words. These are presented to Mapuzugun students which

[15] https://www.duolingo.com/
[16] https://www.lingq.com/en/grammar-resource/
[17] https://www.learnalanguage.com/
[18] https://www.omniglot.com/

reveal promising results. Revita (Katinskaia et al., 2017) automatically create exercises for several endangered languages within the Russian Federation. Specifically, they use morphological analyzers to construct *cloze*-test questions which requires readers to provide the correct surface form of the missing word in a text

**LLMs for Learning** With the advent of Large Language Models (LLMs), many research and commercial applications are exploring their use for language learning, especially for retrieving examples of word collocations or creative writing samples. For example, DuoLingo Max[19] use GPT-4 (OpenAI, 2023) for conversation practice across different scenarios (e.g. going on a vacation versus ordering in a restaurant), which is a useful feature for learning real-world language use. However, this feature is only available for learning high-resource languages of French and Spanish for English speakers. As more languages are added to LLMs, such automatic features can be leveraged across languages and speakers. Additionally, our main focus is to extract interpretable patterns for understanding complex grammatical aspects, which currently is not straightforward to extract from LLMs.

# 8 Next Steps

In this work, we have explored *one* combination of NLP techniques and corpus linguistics to assist in language education. The perception study shows that teachers do find the selected grammar points relevant and interesting, which highlights the importance of a collaborative design; however, all note that the content is more suitable for advanced learners. Although in the current state, the materials cannot be used as-is but the teachers find this overall effort very promising, as this tool can be applied to a corpus of their choice, which is more suited for the learning requirements. Among the different features, teachers find the illustrative examples to be most useful, especially for understanding the non-dominant linguistic behaviors or exceptions to general rules. Additionally, the tool has the capability to extract numerous example usages which the teachers noted as a big plus, as it can provide a starting point for them to build upon rather than them having to find examples manually.

[19]

## 9 Limitations

Currently, a major limitation of the tool, as noted by the teachers, is that the content is not organized by learner age/experience. A next step would be to invite teachers to organize the content by each level, taking the learner incrementally through the complexities of language. For beginner learners, language properties are built through engaging stories with little use of formal grammar terms. Therefore, using simpler meta-language to explain the grammar points and including engaging content would be a worthwhile addition. Even for the teachers, the materials took some time getting used to, especially the formal linguistic terms, therefore, in addition to simplifying the language, educating the teachers in the tool format will also be necessary for effective learning. We hope that our work drives more such practical research in language education where we consult the real users (teachers in our work) to better understand what they need and work with them in collaboration.

## 10 Ethical Considerations

We acknowledge that there are several ethical considerations to keep in mind while creating content or tools that will be directly used by human learners. Since currently we use public corpora to extract the learning materials, they may contain unwanted bias or age-inappropriate language or even culturally-insensitive materials. That is why we work in close collaboration with the respective educators in this work.

## Acknowledgements

We are grateful to Charles Perfetti and Lin Chen from the University of Pittsburgh, for the illuminating discussions on L2 acquisition. We thank all teachers, without whom this work would not have been possible or meaningful. Specifically, we are grateful to Kannada Academy teachers– Arun Sampath, Ashwin Sheshadri, Manasa Kashi, Mukta Hendi, Sunita Sundaresh, Aravind Gangaiah, Shashi Basavaraju, Madhu Rangappagowda, Gayathri Hebbar, Shruthi A, Naina Sharma, Gowri Gudi and P Tantry. We are also grateful to the Marathi Vidyalay, New Jersey teachers– Sudhir Ambekar, Aparna Potdar, Sujata Kulkarni, Varsha Joshi and Archana Kakirde and the Marathi Shala teacher from Pittsburgh– Pranati Talnikar. We also thank Sunanda Tumne, Komal Chaukkar, and Sona Bhide of the Bruhan Maharashtra Mandal (BMM) Shala for providing initial feedback on the interface. We also thank Pruthwik Mishra and Dipti Misra from IIIT-Hyderabad for sharing the Marathi and Kannada treebanks for training the parser. This work is sponsored by the Waibel Presidential Fellowship and the National Science Foundation under grants 1761548, 2125466 and 2109578 . This research was performed under protocol number 2018-00000208 approved by the Carnegie Mellon University Institutional Review Board. Antonios Anastasopoulos is supported by NSF grant IIS-2327143.

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

# A  Appendix

## A.1  Learning a Parser for Marathi and Kannada

As mentioned in the main text, we train our own parser for both Marathi and Kannada to get the POS tags, dependency parses and lemmas. To train a parser for Marathi and Kannada, we use the training data collected by IIIT-Hyderabad[20], which is annotated in the Paninian Grammar Framework (Bhat et al., 2017). However, UDIFY requires training data in the UD annotation scheme (McDonald et al., 2013), so we follow Tandon et al. (2016) to convert between the two formats to obtain POS tags, lemmatization and morphological analysis. However, this converted data does not have dependency information. To obtain dependency data, we train UDIFY in a related language (Hindi) and apply it directly to the converted data above.[21] We then train a new model on this converted data and augment it with the Hindi data, and apply the resulting model on the 4 million Marathi and Kannada raw sentences. The performance of the resulting parser is seen in Table 4.

## A.2  Model for Extracting Learning Material

To extract descriptions in human-readable format, we follow Chaudhary et al. (2022) and Chaudhary et al. (2021) and learn interpretable models such as decision tree (Quinlan, 1986) and SVM as they are conducive to interpretation.

For the grammar aspects of Agreement and Word order, we use XGBOOST to learn a decision tree for each language and setting separately, with the following hyperparameters: learning-rate:0.1, n-estimators:1, subsample:0.8, colsample-bytree:0.8, objective: multi:softprob. We perform a grid-search over two criterion, namely, gini and entropy and depths ranging from 3–20, and select the best performing tree based on the validation set. However, to keep the rules concise, we limit the tree max-depth to 10 and as find that balances the model performance while keeping the number of rules we derive from the trees also concise. We use the standard train/dev/test splits as provided

---

with the original treebanks and report all results in Table 5 The running time of the model is approximately 2-5 mins.

After learning a decision tree, we extract rules from each leaf. However, given that there could be spurious correlations that led to a leaf, simply using the majority label of a leaf as the grammar rule would be incorrect. Therefore, we apply a statistical threshold, as outlined in (Chaudhary et al., 2022) to re-label each leaf. We design two hypothesis, a null hypothesis $H_0$ and a hypothesis to be tested $H_1$, upon which we apply the the chi-squared goodness of fit test where we compute the expected probability distribution for $H_0$ considering a uniform distribution. Below, we define the $H_0$ and $H_1$ for the grammar aspects:

**Morphological Agreement** : The task is formulated as – given a head (e.g. a verb) and dependent (e.g. a noun) in a syntactic relation, when is the agreement for a morphological attribute (e.g. gender) required. This is formulated as a binary classification task, where label is 1 if the values of the head and dependent for the morphological attribute match (e.g. gender = feminine) and 0 otherwise. To extract rules for agreement, we consider those leaves where the majority label is 1. However, simply relying on the majority label could be misleading, as it might be an artifact of any spurious correlations in the training data, we apply the statistical threshold to filter such leaves. Particularly, the null hypothesis $H_0$ states that each leaf denotes chance-agreement i.e. any observed agreement, say for gender between the dependent and its head, is not *required* rather is purely an artifact of the dataset, while $H_1$ states that the leaf being considered denotes required-agreement. If the observed example distribution under a leaf is deemed to be statistically significant when compared to an expected empirical distribution (computed over the training data), we can reject $H_0$ and accept $H_1$.

**Word Order**   The task is formulated as – given a head (e.g. verb) and its dependent (e.g. subject nouns), when is the head before or after its dependent. Similar to above, we design $H_0$ as both the labels i.e. before and after are equally likely, and $H_1$ that the leaf takes the label dominant under that leaf. Leaves that pass the statistical threshold are assigned the dominant label and syntactic/lexical/morphological features that lead up to the leaf form the rule.

---

[20]https://ltrc.iiit.ac.in/showfile.php?filename=downloads/kolhi/

[21]Hindi and Marathi both belong to the same IndoAryan language family and share vocabulary, grammar and even script. Although Kannada belongs to the Dravidian language family, it is still related to Hindi via Sanskrit on which all (Hindi, Marathi and Kannada) are based on.

For extracting rules for suffix usage and word usage, we follow Chaudhary et al. (2021) and use a SVM classifier. The respective tasks are formulated as follows:

**Suffix Usage** The task is formulated as – given a suffix (e.g. -laa) determine the conditions under which this suffix is observed.

**Word Usage** The task is formulated as – given different target language word translations (e.g. 'bhaat' vs 'tandul' for rice) , determine the conditions under which a particular translation is used.

Both these tasks are formulated as multi-class classification tasks, and since Chaudhary et al. (2021) find SVMs to outperform decision trees, we follow their same setup. Specifically, we use the LinearSVM model from sklearn (Fabian, 2011) and perform a grid search over the hyperparameters: C = [0.001, 0.01], class weight =['balanced', None]. We select top-20 features for each word to extract the rules. Furthermore, all rules are formatted using human-readable templates, as shown in Table 2 of Chaudhary et al. (2021).

### A.3 Perception Survey

In Table 3 we present the questions posed to the teachers for assessing the extracted learning material. Consent of all subjects was collected before the study, questions regarding personal information such as name, age, gender, were made optional and all results have been aggregated and presented without revealing individual details.

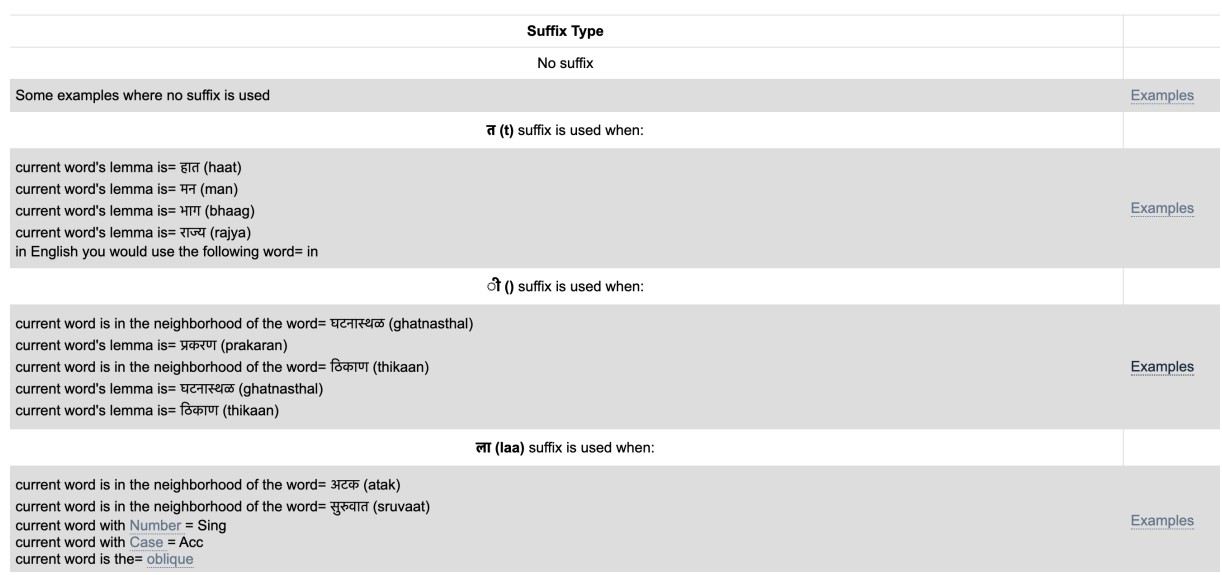

| Suffix Type | |
|---|---|
| No suffix | |
| Some examples where no suffix is used | Examples |
| **त (t)** suffix is used when: | |
| current word's lemma is= हात (haat)
current word's lemma is= मन (man)
current word's lemma is= भाग (bhaag)
current word's lemma is= राज्य (rajya)
in English you would use the following word= in | Examples |
| **ौ ()** suffix is used when: | |
| current word is in the neighborhood of the word= घटनास्थळ (ghatnasthal)
current word's lemma is= प्रकरण (prakaran)
current word is in the neighborhood of the word= ठिकाण (thikaan)
current word's lemma is= घटनास्थळ (ghatnasthal)
current word's lemma is= ठिकाण (thikaan) | Examples |
| **ला (laa)** suffix is used when: | |
| current word is in the neighborhood of the word= अटक (atak)
current word is in the neighborhood of the word= सुरुवात (sruvaat)
current word with Number = Sing
current word with Case = Acc
current word is the= oblique | Examples |

Figure 2: Marathi suffixes for nouns with their usages.

For reference, we also show the corresponding English translation for each example immediately below it

Examples: The **word with suffix च्या** is denoted by \*\*\*

1 या (yaa) हल्ल्यात (hallyaat) \*\*\*इमारतीच्या\*\*\* (imarthia ) खिडक्यांच्या (khidkyanchya) काचा (kaacha) देखील (dekhil) फुटल्या (footlya)

2 the blast also smashed some windows of \*\*\*the\*\*\* building

3 मृतांमध्ये (mritanmadhye) एका (eka) सहा (saha) \*\*\*महिन्यांच्या\*\*\* (mahinyanchya ) बाळाचा (bala) देखील (dekhil) समावेश (samavesh) आहे (aahe)

4 the dead include an \*\*\*eightmonthold\*\*\* child

Figure 3: Illustrative examples extracted for suffix usage. Each example also has Marathi transliteration as well as the English translation to help learners.

Vocabulary covering **nouns (n), verbs (v), adjectives (a), adverbs (r)**

Search for a word (e.g. rice)

| Type | |
|---|---|
| food (n) | chocolate -- चॉकलेट (chocolate), Examples
pepper -- मीठ (meeth), Examples
sugar -- साखर (saakhar), Examples
fodder -- चारा (chaara), Examples
food -- अन्न (ann), Examples
rice -- तांदूळ (tandul), Examples
nutrient -- अस (as), Examples
liquor -- दारू (daaru), Examples
stock -- शेअर (share), Examples
flour -- पीठ (peeth), Examples
lemon -- लिंब (limb), Examples
chop -- चिर (chir), Examples
produce -- निर्मिती (nirmiti), Examples
vegetable -- भाजी (bhaaji), Examples |
| relationships (n) | sibling -- आहे (aahe), Examples
brother -- भाऊ (bhaau), Examples
dad -- बाबा (baba), Examples |

Figure 4: Marathi lemmas organized by basic categories. Each lemma contains a link to illustrative examples which shows its usage in full sentential context, with its English translations.

Adjectives with synonyms and antonyms

Search for a word (e.g. rice)

| English Word | Definition | Marathi Word | Synonyms | Antonyms |
|---|---|---|---|---|
| first | the first or highest in an ordering or series | पहिला (pahila), Examples | | |
| social | a party of people assembled to promote sociability and communal activity | सोशल (soshal), Examples | | |
| new | not of long duration; having just (or relatively recently) come into being or been made or acquired or discovered | नवा (nava), Examples | ताजा (taaja), Examples | |
| important | of great significance or value | महत्व (mahatva), Examples | मोठा (mothaa), Examples | |
| private | an enlisted man of the lowest rank in the Army or Marines | खासगी (khaasgi), Examples | | सार्वजनिक (saarvajanik), Examples |

Figure 5: Marathi adjectives with their definitions, synonyms and antonyms.

| Type | Question | Answer Choices |
|------|----------|----------------|
| Teacher Background | Name (Optional)
Age (Optional)
Gender (Optional)
How long have you been teaching Kannada?
What level of learners do you teach?
Have you used computer-based tools for your teaching? | |
| Relevance | **1.** What percentage of the materials presented in the tool cover existing curriculum? | 0-100% |
| Utility | **2.** How likely are you inclined to use this tool in your teaching? | 3: Highly likely
2: Likely
1: Not likely |
| | **2.1.** If likely, for what purpose do you foresee this being used?
(multiple answers can be selected): | a. For lesson preparation, knowledge
b. For evaluating students
c. Present to the students for self-exploration
d. Other (please specify the reason) |
| | **2.2.** If likely, what aspects would you use:

(multiple answers can be selected): | a. The general concept introduced by the material
b. The rules described in the tool
c. Illustrative examples that accompany the rule
d. Other (please specify the reason) |
| | **2.3.** if NOT likely, why?

(multiple answers can be selected): | a. material outside the scope
b. material unclear and needs improvement
c. material already covered by existing curriculum
d. Other (please specify the reason) |
| Presentation | **3.** How did you find the tool? | 3. Very easy to use and navigate
2. Somewhat easy to use, but took some time to get used to
1. Difficult to use |
| Feedback | **4.1** What did you like about the tool?
**4.2** What did you not like about the tool?
**4.3** What would you like to improve in the tool? | |

Table 3: Usability study: Questions posed to the in-service teachers for evaluating the learning materials on relevance, utility and presentation. This set of questions is asked for each grammar concept.

| Language | Train / Dev / Test | POS | Morphological Analysis | Lemmatization |
|----------|--------------------|-----|------------------------|---------------|
| Marathi (PAN) | 11518 / 1490 / 1503 | 85.9 | 70.1 | 82.2 |
| Kannada (PAN) | 7244 / 1956 /40 | 90.3 | 79.3 | 90.6 |

Table 4: We evaluate the parser performance on the respective test sets of the Paninian treebanks (PAN). Since we only have gold annotations for POS, morphological analyses and lemmas, we only report results for those. In the second column, we report the number of train/dev/test sentences used in the UDIFY parser.

| Grammar Concept | Type | Kannada | | Marathi | |
|---|---|---|---|---|---|
| | | **AutoLex** | baseline | **AutoLex** | baseline |
| Word Order | subject-verb | **97.02** (7) | 96.97 | **97.8** (13) | 97.7 |
| | object-verb | **99.11** (6) | 99.06 | **97.89** (13) | 96.78 |
| | numeral-noun | **98.63** (5) | 98.36 | 99.54 (1) | 99.54 |
| | adjective-noun | 99.92 (1) | 99.92 | - | - |
| | noun-adposition | 99.14 (2) | 99.14 | - | - |
| Agreement | Gender | **71.87** (31) | 65.69 | 61.11 (84) | **81.44** |
| | Person | 24.73 (29) | **25.16** | - | - |
| Suffix Usage | NST | **91.58** (13) | 50 | **90.44** (1) | 93.77 |
| | NUM | **85.2** (5) | 82.63 | 85.91 (1) | **93.61** |
| | NOUN | **78.61** (19) | 39 | **70.23** (13) | 67.8 |
| | PRON | **87.13** (15) | 58.03 | **75.66** (10) | 65.07 |
| | PART | **94.73** (13) | 89.35 | **90.58** (4) | 76.77 |
| | ADJ | **87.74** (14) | 66.82 | **87.55** (4) | 83.83 |
| | VERB | **63.19** (14) | 30.52 | **78.44** (10) | 65.87 |
| | PROPN | **74.57** (16) | 46.68 | 65.6 (9) | **71.19** |
| | SCONJ | **96.85** (9) | 64.6 | **97.59** (5) | 86.9 |
| | DET | **99.53** (4) | 61.83 | **83.91** (2) | 81.71 |
| | AUX | **76.92** (5) | 38.46 | **92.8** (6) | 81.57 |
| | ADV | **75.19** (12) | 37.27 | **86.84** (9) | 65.89 |
| | ADP | **93.55** (6) | 76.43 | **97.12** (6) | 67.63 |
| Vocabulary | Semantic Subdivisions | **68.68** (385) | 58.48 | **70.58** (285) | 56.26 |

Table 5: Automated evaluation results for learning materials extracted for each grammar concept. Number in the bracket denotes the number of rules extracted for word order, agreement and suffix usage, while for vocabulary it denotes the number of word pairs that show fine-grained distinctions.