# OpenReview forum: "Teacher Perception of Automatically Extracted Grammar Concepts for L2 Language Learning"
_EMNLP/2023/Conference — EMNLP 2023 Findings_

### Official Review · Reviewer_61GB · 2023-08-01

**Soundness:** 3

**Excitement:**

4: Strong: This paper deepens the understanding of some phenomenon or lowers the barriers to an existing research direction.

**Justification For Ethical Concerns:**

I find no  ethical concerns with this study.

**Missing References:**

N/A

**Paper Topic And Main Contributions:**

This paper deals with automated methods to prepare educational materials for L2 learners of the languages Kannada and Marathi (learners whose L1 is English).
Specifically, the author utilize the previously published AutoLex system. Using some NLP-preprocessed corpora, the overall system produces educational materials (rules and examples) for vocabulary, morphology, grammar and senses.
The study includes reports on interviews with teachers and their ratings concerning the utility of such system for language instructors/teachers.

**Questions For The Authors:**

The system heavily depends on AutoLex (a previously published work),
and the manuscript is difficult to understand for those who are not familiar with AutoLex.
I suggest to add to the manuscript a short paragraph to describe what AutoLex does and how.

**Reasons To Accept:**

This is a very interesting work and quite important, in the following aspects.
First, it utilizes NLP approaches for preparation of educational materials, which is a very practical but often neglected aspect of NLP.
Second, it does so in the languages Kannada and Marathi, which generally lack extensive NLP resources.
Third, it is just refreshing to have some papers that are not about Deep Learning.

**Reasons To Reject:**

The described system already works, but it seems to be still in development,
and some aspects of it are not yet 'fully' developed (in some cases the error rate is too high).
Also, it seems the applicability is still in development - it is not clear which parts of the system might be suitable for teaching of beginner-learners, and which are more suitable for advanced learners, and how to tune such aspects.
Obviously much more work needs to be done along those lines.
However even the current version (as described in the manuscript) is very promising and deserves publication.



**Reproducibility:**

3: Could reproduce the results with some difficulty. The settings of parameters are underspecified or subjectively determined; the training/evaluation data are not widely available.

**Reviewer Confidence:**

3: Pretty sure, but there's a chance I missed something. Although I have a good feel for this area in general, I did not carefully check the paper's details, e.g., the math, experimental design, or novelty.

**Typos Grammar Style And Presentation Improvements:**

Line 396: what is "popular adjectives" ?  Did you mean most frequent ones based on a corpus?

---

> ### Author Rebuttal · Authors · 2023-08-27
>
> We would like to thank the reviewer for their encouraging words on the motivation of our work and for their valuable feedback. Please find our response below:
>
> **The described system already works, but it seems to be still in development, not clear which parts of the system might be suitable for teaching of beginner-learners, and which for advanced learners, and how to tune such aspects**
> >We completely agree that this is still in development. The main goal of our work was to investigate whether even using such automatic methods can produce pedagogical materials especially for languages which lack pedagogy resources and we believe that through our work the answer is “yes”. We hope this drives more such practical research in language education where we consult the real users (teachers in our work) to better understand what they need and work with them in collaboration.
> Its indeed an interesting research problem on how to better curate the materials according to learner levels, especially when the schools do not use global standards such as CEFR but instead create their own. One option could be to run our approach on raw text which is more suited to a particular level, but this is not enough as we would need to know what concepts to be introduced at each level as well.
>
> **I suggest to add to the manuscript a short paragraph to describe what AutoLex does and how.**
> >Thanks for pointing it out, we will add more details on AutoLEX to help the reader better understand.
>
> **Line 396: what is "popular adjectives" ? Did you mean most frequent ones based on a corpus?**
> >We automatically identified those cross-lingual word pairs that were frequently aligned with each other in our corpus, identified using POS which of those were adjectives and used WordNet to identify the antonyms/synonyms of their English counterparts. We will clarify this in the paper.

---

### Official Review · Reviewer_r9y4 · 2023-08-01

**Soundness:** 4

**Excitement:**

4: Strong: This paper deepens the understanding of some phenomenon or lowers the barriers to an existing research direction.

**Paper Topic And Main Contributions:**

The authors propose methods for automatically extracting learning materials for two Indian languages, Kannada and Marathi, which have low resources for second-language learners. These learning materials include teachable grammar points for word order and agreement, suffix usage, vocabulary, and general information.

The work is built upon AutoLEX (Chaudhary et al., 2022), which formulates a linguistic question into an NLP prediction task. Models are then trained for this prediction task by extracting relevant features from text corpora, such as POS tagging and dependency parsing, which are known to influence the targeted task. Specific examples can also be presented to the learner.

The authors use AutoLEX for word order and agreement but adapt it for the rest of above mentioned teachable grammar points.
The paper includes a very thorough human evaluation, which was based on a survey completed by multiple language educators.
The proposed approach is simple and interesting, and targets low-resource languages, however, I see mixed results in the manual evaluations.

**Questions For The Authors:**

- Do you think this material can be used directly by a learner? If so, it would have been nice to include some learners in your analysis to see how much they find the material helpful as someone who is trying to learn a second language.
- In A2, there are not many details for the specifics of the model you trained. Please include that.
- The survey covers relevance and utility, but not accuracy/correctness. I think this should have been part of the survey since extractions are automatic.

**Reasons To Accept:**

- authors study an interesting and useful task
- their approach is simple, straightforward, and explainable.
- I like the fact that they are able to include specific examples and exceptions for extracted rules.
- they target two low-resource languages
- they include a very thorough human evaluation (although I have questions about how they designed it)

**Reasons To Reject:**

- Authors are mainly following the approach proposed by Chaudhary et al., 2022. So I don't find the approach very novel.
- Based on the human analysis, it doesn't seem like the method works that well. For Kannada-relevance was 45-60% and Marathi-relevance was 10-15%. If extracted material is not relevant, then how helpful could it be? Also, there is not much discussion on the correctness of the presented material.

**Reproducibility:**

4: Could mostly reproduce the results, but there may be some variation because of sample variance or minor variations in their interpretation of the protocol or method.

**Reviewer Confidence:**

4: Quite sure. I tried to check the important points carefully. It's unlikely, though conceivable, that I missed something that should affect my ratings.

**Typos Grammar Style And Presentation Improvements:**

- related work is in the appendix which is a bit unusual IMO.
- footnote 17 is missing
- Model section should include more details so that the reader doesn't have to look into Chaudhary et al. 2021, 2022

---

> ### Author Rebuttal · Authors · 2023-08-27
>
> We thank the reviewer for their detailed and careful review, and providing suggestions for presentation improvement. Please find our response below.
>
> **Authors are mainly following the approach proposed by Chaudhary et al., 2022. So I don't find the approach very novel.**
> >We would like to clarify the novelty of our work, our main contribution is to explore how NLP methods can play a role in language pedagogy especially for languages which lack resources. This entails a) identifying what learning materials to extract, for which we collaborate with teachers, b) extracting the materials from raw text, c) conducting a principled study with in-service teachers to assess its real-world utility.
> The approach proposed by Chaudhary et al., 2022.  is just used for b), and could be replaced by some other model which can extract human-readable explanations. Furthermore, we show how to make the approach proposed by Chaudhary et al., 2022 work with true low-resource languages that lack syntactic information (eg. POS tags, dependency parses, morphology) that Chaudhary et al., 2022. assume exists in a particular UD schema. Moreover, we design our teaching points by including real users (teachers in our case) in a collaborative design process and further evaluate its practical utility with a large human study.
>  **In sum, the main novelty of our work is a comprehensive experimental study with actual teachers of low-resource languages, but in order to make this study possible we also made a few novel methodological improvements as well. We will make this more clear in the camera-ready.**
>
> **Based on the human analysis, it doesn't seem like the method works that well. For Kannada-relevance was 45-60% and Marathi-relevance was 10-15%. If extracted material is not relevant, then how helpful could it be?**
> >We do agree that currently the materials extracted are not highly relevant but as we also note in L529 and L398, the only sufficiently large publicly available Kannada and Marathi corpora comprised mostly of domains such as news, Wikipedia, talks, religious text, movies which are not particularly suitable for pedagogy. But despite this domain-shift, the teachers do find that the proposed technique has potential and nearly 80% Kannada teachers and 60% Marathi teachers said they were likely to use it in practice, more details on utility per grammar point is in Section 6. The teachers also noted that in L340 that  if the underlying corpora was more learner oriented then the resulting materials would be even more useful. Given that this is the first study of its type (to our knowledge), we also believe there is value in pointing out some of the difficulties in applying these methods, so that they may be further resolved in future work.
>
>
> **Also, there is not much discussion on the correctness of the presented material**
> >Regarding correctness of the material – in Section 5 we conduct a limited human evaluation where the Kannada experts evaluate some questions about word order, suffix usage and vocabulary. Given that its not feasible to manually evaluate all the grammar points, which comprises >200 rules (Table 5), we also conduct an automatic evaluation as shown in Table 5 which shows that the model outperforms the respective baselines. If you have other suggestions on how to better present the results on correctness we would be happy to consider modifications for the camera ready.
>
> **Do you think this material can be used directly by a learner? If so, it would have been nice to include some learners in your analysis to see how much they find the material helpful as someone who is trying to learn a second language.**
> >The materials in the current state cannot be directly used by a learner, because currently the materials are designed for the teachers. We do this for two reasons: a) the teachers are best suited to decide how to present the materials to the learners, and b) given that the materials are automatically extracted over public data, the examples derived from the dataset could be age-inappropriate or could contain offensive content, hence we do not want to expose learners to unverified materials.
> Given that our focus was on teacher assistance, for the materials to be directly useful for a learner, they need to be categorised by learner level, verified by experts and the language modified to age-appropriate, which is not a straightforward task and we leave it for future work. Interestingly, one of the Kannada teachers did use some of our extracted materials to teach suffix usage in their lesson, as shown in L552. The teacher did note that they liked our suffix usage material, but that the examples for those were more suited for written language learning rather than spoken language.
>
> **The survey covers relevance and utility, but not accuracy/correctness. I think this should have been part of the survey since extractions are automatic.**
> >Thank you for the suggestion, we do conduct a limited human evaluation of the materials with two teachers in Section 5 accompanied by an automatic evaluation of all rules in Section 4. However, it is not feasible to conduct this evaluation for all grammar rules, which are more than 200 (Table 5), making this a time-consuming task. For reference, the limited evaluation by the two experts itself took almost two weeks.
>
> **Model section should include more details, related work in appendix**
> >Thank you for pointing it out, we will add more details on the modeling approach to avoid any confusion. Due to the brevity of space, we moved the related work to appendix but when we are granted the extra page in camera-ready we could move it back to the main paper.

---

### Official Review · Reviewer_iWaN · 2023-08-06

**Soundness:** 3

**Excitement:**

3: Ambivalent: It has merits (e.g., it reports state-of-the-art results, the idea is nice), but there are key weaknesses (e.g., it describes incremental work), and it can significantly benefit from another round of revision. However, I won't object to accepting it if my co-reviewers champion it.

**Paper Topic And Main Contributions:**

This paper presents a study of the applicability of NLP tools for assisting language teaching. The authors used their tool (AutoLEX) to extract grammar-like rules concerning word order, word agreement, suffixes and vocabulary in Kannada and Marathi languages (both resource-poor languages). After evaluating the extraction, the authors assess the utility of the output in a teaching context. To that end, 12 (Kannada) and 5 (Marathi) non-professional teachers evaluated the results concerning relevance, utility and presentation.

**Questions For The Authors:**

Were the evaluators (Section 4) and teachers (Section 6) remunerated?

Does the assessment made in the perception survey (Table 2) include the items identified as incorrect (Section 5)?

**Reasons To Accept:**

The work partially fills the gap at the interface between NLP and teaching practice.

An end-to-end pipeline for 2 resource-poor languages.

**Reasons To Reject:**

There is a lack of curriculum. The results were considered more suitable for advanced learners and not relevant for beginners. However, it is not clear what is the level definition (or even if both schools use the same definition). In addition, the linguistic features studied are usually more associated with beginner levels (e.g., A1 and A2 in CEFR).

The objective stated is imprecise concerning what has been presented. The authors claim they "explore to what extent can a combination of NLP techniques and corpus linguistics assist language education for languages." However, only a few methods to support language teaching are used.

There is a lack of contextualization about the paper's position on supporting the teacher (e.g., material development) and/or the student (e.g., free exploration and feedback generation).

**Reproducibility:**

4: Could mostly reproduce the results, but there may be some variation because of sample variance or minor variations in their interpretation of the protocol or method.

**Reviewer Confidence:**

3: Pretty sure, but there's a chance I missed something. Although I have a good feel for this area in general, I did not carefully check the paper's details, e.g., the math, experimental design, or novelty.

---

> ### Author Rebuttal · Authors · 2023-08-27
>
> We thank the reviewer for their valuable comments and suggestions. We have attempted to clarify some in this response and will work towards completely considering all in the camera-ready.
>
> **what is the level definition or if both schools used the same definition**
> >Thank you for asking this question! The Kannada and Marathi schools that we collaborated with didn’t use any global standards (e.g. CEFR) as they found them unsuitable for their needs, instead they came up with their own. As described in Section 3.1, the teachers shared the eight textbooks they used upon which we built our study. Given that their textbook levels are not based on CEFR, it is not straightforward to assign a standard level (e.g. A1) to our  materials. Additionally, our goal in this work is to provide the materials to the teachers and not the learners directly, as the teachers are best suited to select the appropriate material for different learners.
>
> **lack of contextualization about paper's position on supporting the teacher (e.g., material development) and/or the student (e.g., free exploration and feedback generation)**
> >Our focus is on supporting the teachers primarily, for which we extract both grammar rules and illustrative examples, the motivation being that examples could assist the teachers in their teaching process (as we indeed find in Table 3 that these materials could be useful for lesson prep). Interestingly, the teachers found these illustrative examples could also be given to the students for their own exploration. Moreover, since these materials are automatically extracted and could be prone to errors or may contain material which is age-inappropriate, we chose to share these materials with the teachers who are best suited to decide how to use them. We will add this clarification in our camera-ready.
>
> **Were the evaluators (Section 4) and teachers (Section 6) remunerated?**
> >No, there was no remuneration, all teachers volunteered.
>
> **Does the assessment made in the perception survey (Table 2) include the items identified as incorrect (Section 5)?**
> >Yes, we did not remove those items as we wanted to understand how the materials, as directly obtained automatically without significant human intervention, are perceived when presented as-is. This is close to the real world setting where human evaluation of each grammar point is not feasible. Thanks for asking this question, we will add this explanation in the camera-ready.
>
> **The objective stated is imprecise concerning what has been presented. The authors claim they "explore to what extent can a combination of NLP techniques and corpus linguistics assist language education for languages." However, only a few methods to support language teaching are used.**
> >Thank you for pointing it out, we would like to clarify that by this statement we meant that in this work, we want to explore *one* such combination for automatically extracting language learning materials. Although, comparing different extraction methods would be an interesting future direction. We will clarify this in the paper.

---

### Meta-Review · Area_Chair_F27o · 2023-09-18

**Recommendation:** 4

**Metareview:**

The paper investigates the applicability of NLP for creating learning materials for second language learning; evaluation with language educators from North America.

There is consensus among the reviewers that this paper presents a valuable contribution. Reviewers especially appreciate the importance and practical application of the task (education), the focus on two under-resourced languages, and the thorough human evaluation. While reviewers (and the authors) acknowledge some mixed results/limitations of the work (e.g., there is work to do before the approach is fully usable/applicable), the reviewers nevertheless find the current work promising.

---

### Decision · Program_Chairs · 2023-10-07

**Decision:**

Accept-Findings

**Comment:**

The paper investigates the applicability of NLP for creating learning materials for second language learning; evaluation with language educators from North America.

There is consensus among the reviewers that this paper presents a valuable contribution. Reviewers especially appreciate the importance and practical application of the task (education), the focus on two under-resourced languages, and the thorough human evaluation. While reviewers (and the authors) acknowledge some mixed results/limitations of the work (e.g., there is work to do before the approach is fully usable/applicable), the reviewers nevertheless find the current work promising.